# Signatures in SARS-CoV-2 spike protein conferring escape to neutralizing antibodies

**Marta Alenquer[1], Filipe Ferreira[1☯], Diana Lousa[2☯], Mariana Valério[2], Mónica Medina-Lopes[1], Marie-Louise Bergman[3], Juliana Gonçalves[4], Jocelyne Demengeot[3], Ricardo B. Leite[5], Jingtao Lilue[6], Zemin Ning[7], Carlos Penha-Gonçalves[8], Helena Soares[4], Cláudio M. Soares[2], Maria João Amorim[1]***

1 Cell Biology of Viral Infection Lab, Instituto Gulbenkian de Ciência; Oeiras, Portugal, 2 ITQB NOVA, Instituto de Tecnologia Química e Biológica António Xavier, Universidade Nova de Lisboa; Oeiras, Portugal, 3 Lymphocyte Physiology Lab, Instituto Gulbenkian de Ciência; Oeiras, Portugal, 4 CEDOC NOVA, Centro de Estudos de Doenças Crónicas, Nova Medical School, Universidade Nova de Lisboa; Lisboa, Portugal, 5 Genomics Unit, Instituto Gulbenkian de Ciência; Oeiras, Portugal, 6 Bioinformatics Unit, Instituto Gulbenkian de Ciência; Oeiras, Portugal, 7 The Wellcome Trust Sanger Institute; Hinxton, United Kingdom, 8 Disease Genetics Lab, Instituto Gulbenkian de Ciência; Oeiras, Portugal

☯ These authors contributed equally to this work.
* mjamorim@igc.gulbenkian.pt

**Data Availability Statement:** All relevant data are within the manuscript and its Supporting Information files.

## Abstract

Understanding SARS-CoV-2 evolution and host immunity is critical to control COVID-19 pandemics. At the core is an arms-race between SARS-CoV-2 antibody and angiotensin-converting enzyme 2 (ACE2) recognition, a function of the viral protein spike. Mutations in spike impacting antibody and/or ACE2 binding are appearing worldwide, imposing the need to monitor SARS-CoV2 evolution and dynamics in the population. Determining signatures in SARS-CoV-2 that render the virus resistant to neutralizing antibodies is critical. We engineered 25 spike-pseudotyped lentiviruses containing individual and combined mutations in the spike protein, including all defining mutations in the variants of concern, to identify the effect of single and synergic amino acid substitutions in promoting immune escape. We confirmed that E484K evades antibody neutralization elicited by infection or vaccination, a capacity augmented when complemented by K417N and N501Y mutations. *In silico* analysis provided an explanation for E484K immune evasion. E484 frequently engages in interactions with antibodies but not with ACE2. Importantly, we identified a novel amino acid of concern, S494, which shares a similar pattern. Using the already circulating mutation S494P, we found that it reduces antibody neutralization of convalescent and post-immunization sera, particularly when combined with E484K and with mutations able to increase binding to ACE2, such as N501Y. Our analysis of synergic mutations provides a signature for hotspots for immune evasion and for targets of therapies, vaccines and diagnostics.

## Author summary

For a SARS-CoV-2 virion to enter a cell, the spike protein displayed at its surface must be recognized by the host ACE2 receptor. Serum neutralizing antibodies, shown to develop

**Funding:** This project is supported by the grants PTDC/CCI-BIO/28200/2017 awarded to DF; FCT RESEARCH4COVID 19 (Refs 580 and 644) awarded by the Fundação para a Ciência e a Tecnologia (FCT, Portugal, https://www.fct.pt/index.phtml.en) to MJA and also by COVID-19 emergency funds 2020 from Fundação Calouste Gulbenkian - Instituto Gulbenkian de Ciência (FCG-IGC, Portugal, https://gulbenkian.pt/ciencia/) to MJA. The work of JD benefited from COVID-19 emergency funds 2020 from Fundação Gulbenkian de Ciência and from Câmara Municipal de Oeiras (https://www.oeiras.pt). The work of HS was supported by ESCMID (https://www.escmid.org) and by Gilead Génese (PGG/009/2017, https://gileadgenese.pt) grants. MJA is funded by FCT 2020.02373.CEECIND; FF is supported by the FCT grant PTDC/BIA-CEL/32211/2017; JG and HS are supported by Fundação para a Ciência e Tecnologia (FCT) through PD/BD/128343/2017 and CEECIND/01049/2020.

**Competing interests:** The authors have declared that no competing interests exist.

upon natural SARS-CoV-2 infection or vaccination, bind spike protein preventing the recognition by ACE2 and, consequently, infection. However, SARS-CoV-2 virus is constantly evolving, and can acquire mutations in spike that render this protein resistant to neutralizing antibodies and make vaccines ineffective. In this paper, we tested how single and a combination of mutations naturally occurring in spike, including in variants of concern, would synergize to affect antibody neutralizing capacity. We then integrated these findings with *in silico* analyses of amino acids binding to ACE2 and antibodies, and distributed them in a grid as amino acids important for binding to ACE2 or antibodies, or both. We found that changes in amino such as E484 and S494, which frequently interact with antibodies but not with ACE2, promptly evolve immune escape mutants, elicited by infection or vaccination, if the mutation severely alters the binding specificity of the antibody. Our work also suggests that the combination of these mutations with others promoting ACE2 binding, such as N501Y, increases their ability to escape neutralizing-antibody responses.

## Introduction

Severe acute respiratory syndrome coronavirus–2 (SARS-CoV-2) is the virus responsible for the pandemic of coronavirus disease 2019 (COVID-19) [1] that has caused more than 164 million infections and provoked the death of over 3 million people (as of April 18, 2021). A pertinent question is whether viral evolution will permit escaping immunity developed by natural infection or by vaccination. The answer to this multi-layered and complex question depends on the type, duration and heterogeneity of the selective pressures imposed by the host and by the environment to the virus, but also on the rate and phenotypic impact of the mutations the virus acquires. The central question is whether the virus can select mutations that escape host immunity while remaining efficient to replicate in the host [2–4]. Identifying immune evasion signatures in the virus is critical for preparedness of future interventions to control COVID-19.

SARS-CoV-2 is an enveloped virus, characterized by displaying spike proteins at the surface [5]. Spike is critical for viral entry [6] and is the primary target of vaccines and therapeutic strategies, as this protein is the immunodominant target for antibodies [7–11]. Spike binds to ACE2 displayed at the host cell surface and, upon activation by proteolytic cleavage, mediates membrane fusion of viral and host membranes to deliver virion contents inside the cell. Proteolytic cleavage happens at two sites in spike (S1/S2 and S2') in a process that starts during viral egress with cleavage of S1/S2 site (by furin) and terminates with cleavage of the S2' site either at the cell surface by TMPRSS2 or inside lysosomes by, for example, cathepsin L [6]. Interestingly, the spike protein oscillates between distinct conformations to recognize and bind different host factors [12,13]. In the prefusion state, the receptor binding domain (RBD) alternates between open ('up') and closed ('down') conformations [5,14]. For binding to the ACE2 receptor, the RBD is transiently exposed in the 'up' conformation. However, most potent neutralizing antibodies elicited by natural infection [9,11,15] and by vaccination [7,10] bind spike in the closed ('down') conformation. Understanding how each amino acid of spike dynamically interacts with either antibodies or ACE2 receptor may reveal the residues that are more prone to suffer mutations driving immunological escape without affecting viral entry.

Viral mutations may affect host-pathogen interactions in many ways: affect viral spread, impact virulence, escape natural or vaccine-induced immunity, evade therapies or detection by diagnostic tests, and change host species range [16,17]. Therefore, it is critical to survey

circulating variants and assess their impact in the progression of SARS-CoV-2 dynamics in the population, in real time. Being a novel virus circulating in the human population, SARS-CoV-2 evolution displayed a mutational pattern of mostly neutral random genetic drift until December 2020. In fact, the D614G mutation in the spike protein was amongst the few epidemiologically significant variants, resulting in increased transmissibility without affecting the severity of the disease [18,19]. However, from the end of 2020, three divergent SARS-CoV-2 lineages evolved into fast-spreading variants that became known as variants of concern: B.1.1.7 [United Kingdom (UK)] [20], B.1.351 [South Africa (SA)] [21] and P.1 (Brazil) [22]. More recently, other lineages were added to this list: B.1.427, B.1.429 [California, United States of America (USA)], B.1.617 or B.1.617.1–3 (India) [20,23,24], as well as several lineages of interest that carry the mutation E484K, including P.2 [25] and B.1.1.7 containing E484K. These variants present alterations in the spike protein that change its properties: increase transmissibility and virulence [26,27] and/or escape immunity developed by natural infection [28,29], therapies [30–32], vaccination [33], detection [34]; and change the host species range [35]. Mutations in the RBD of spike may severely affect viral replication and host immune response since, as explained above, this region is responsible for binding to ACE2 and is immunodominant. Three mutations—D614G, N501Y and L452R - are associated with an increased ACE2 binding affinity in humans and increased viral transmission [18,19,26,36]. Y453F was associated with a mink-to-human adaptation in cluster 5 [37]. L452R and N439K are associated with a modest reduction in antibody-dependent neutralization by immune sera, whilst variants containing E484K display a reduction that is moderate to substantial [38]. Variants, however, contain several mutations combined and some will have synergetic effects. Interestingly, some mutations are convergent whilst others are unique in lineages [39]. For example, in B.1.351 and P.1 lineages, mutations in amino acid residues at positions 18 (L18F) and at position 417 (S417N, or S417T in some P.1 cases) were observed. A two amino acid deletion in positions 69 and 70 of spike is observed in cluster 5 and lineages B.1.1.7, B.1.525 and B.1.258 [40]. The appearance of shared mutations in distinct and rapidly spreading SARS-CoV-2 lineages suggests that these mutations, either alone or in combination, may provide fitness advantage, as recently observed with the acquisition of S417N by B.1.617.2 lineage. In principle, mutations in different domains of spike may affect its function and result in conformational rearrangements of the RBD. Therefore, it is important to understand how synergetic interactions in spike collectively change the RBD.

Serum neutralizing antibodies were shown to develop upon natural SARS-CoV-2 infection [41] or vaccination [42,43], and last at least several months [44–46]. The alarm caused by the emergence of SARS-CoV-2 variants with the potential to escape host immunity acquired through infection or vaccination has been confirmed by a series of reports for lineages B.1.351 and P.1. [28,29,33,47,48]. These raise concerns on whether a reduction in vaccine efficacy could result in re-infections and delay the reduction in mortality caused by circulating SARS-CoV-2. There is, therefore, the urgent need to develop mutation-tolerant vaccines (and biopharmaceuticals) targeting the spike protein. In this sense, it is critical to determine what makes mutations well tolerated for the viral lifecycle whilst efficiently escaping immunity. In this work, we engineered spike-pseudotyped viruses and analyzed individual and combined mutations that convergently appeared in different lineages, over time and across several geographic locations, to determine their synergetic effects on neutralizing-antibody responses. We then used available structures of the complexes spike-antibodies and spike-ACE2 to determine the distance between each amino acid residue in the complex and the frequency of interactions of each amino acid residue of the RBD with either ACE2 or antibodies. We found a moderate reduction in neutralizing potency of sera against SARS-CoV-2 spike-pseudotyped lentivirus containing single mutations at positions 484 (E484K) and 494 (S494P). Interestingly,

the reduction became substantial with the addition of synergetic mutations K417N/N501Y to E484K, or E484K/N501Y to S494P. In addition, we show that the amino acid residues at positions 484 and 494 frequently engage in binding to antibodies but not in binding to the receptor ACE2. Our work suggests that the amino acid residues at the RBD that are more dispensable for binding to ACE2 can more promptly evolve immune escape mutants if the amino acid residue substitution severely alters the binding specificity of the antibody. It also suggests that the combination of these mutations with others that promote ACE2 binding, such as N501Y, increases their ability to escape neutralizing-antibody responses.

## Results

### Construction of spike-pseudotyped viruses containing single or combined mutations in spike

The identification of mutations in SARS-CoV-2 genome of considerable prevalence or that emerged independently over time is important, not only to track viral evolution and dynamics of lineages circulating worldwide, but also to identify recurrent mutations and their effects, and ultimately tailor preventive measures to contain the virus. For this study, we selected only mutations in spike (that emerged between 29[th] of December 2019 and 25[th] of March 2021, S1 Fig) because this protein is responsible for viral recognition of ACE2 receptor [6,49–52] and for inducing neutralizing antibodies in the host [10] and, therefore, mutations in spike may hamper therapies, populational immunity and vaccination-mediated protection [53–55]. Whilst mutations on the RBD may sterically block binding of spike to ACE2, mutations in different spike domains may affect the conformation of the protein thereby impacting antibody recognition. Given this, for this study we selected mutations in spike to include variants of concern and of interest (up to all mutations in lineages B.1.1.7, B.1.351 and P.1; S1 Table), of high prevalence (S1 Fig), or convergent appearance in different lineages over time, as shown in Fig 1B and described in S1 Table. In addition, the mutations in the RBD region are highlighted also in the open ('up') conformation (Fig 1C), showing the conformational changes that each amino acid residue undergoes in this region.

The different spike versions served to engineer lentiviral spike-pseudotyped particles through a three-plasmid approach: a plasmid harboring the lentiviral genome, lacking Gag-Pol and envelope proteins, and encoding a GFP reporter; a Gag-Pol expression plasmid; and a plasmid expressing SARS-CoV-2 spike protein (or VSV-G protein, as control). Cells infected by these lentiviruses express GFP and can be easily quantified with high-throughput analytical methods.

### Neutralization assays on spike-pseudotyped viruses to evaluate synergetic effects of mutations

For the neutralization assay, we developed a 293T cell line expressing the human ACE2 receptor (S2 Fig). The specificity of the assay was assessed using an anti-spike polyclonal antibody and an RBD peptide that competes with the spike protein for viral entry in SARS-CoV-2 spike-pseudotyped but not in VSV G-pseudotyped lentivirus that was used in parallel (S3 Fig).

In this study, we evaluated evasion of host neutralizing antibodies. We characterized the serum antibodies from 12–16 health care workers, infected with SARS-CoV-2 during spring/ summer 2020, for their ability to neutralize our spike-pseudotyped mutant lentiviruses. These sera were tested by ELISA for their anti-spike content and classified as low (endpoint titer up to 1:150), medium (endpoint titer 1:450) and high (endpoint titer from 1:1350) (S2 Table). We also used 4 negative sera as control (one pre-pandemic and 3 contemporary). Representative

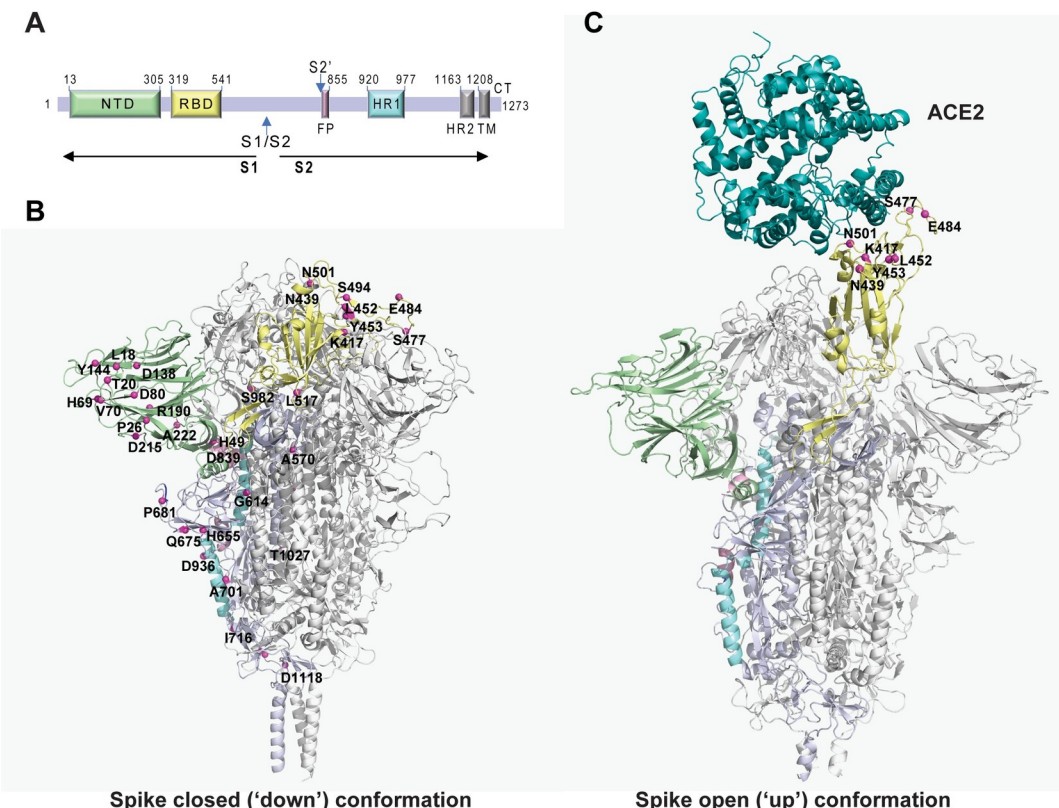

**Fig 1. Location of the mutations in spike tested in this work. (A)** Schematic representation of the SARS-CoV-2 spike primary structure. NTD, N-terminal domain; RBD, receptor-binding domain; S1/S2, S1/S2 protease cleavage site; S2', S2' protease cleavage site; FP, fusion peptide; HR1, heptad repeat 1; HR2, heptad repeat 2; TM, transmembrane domain; CT, cytoplasmic tail. The protease cleavage sites are indicated by arrows. **(B)** The mutations in spike engineered in this study are mapped onto the spike protein surface, using a structure obtained by homology-based modelling with the structure PDB ID 6XR8 as a template. Mutations are highlighted using magenta spheres mapped on one of the monomers, which is represented with different colors to highlight important regions: NTD (green), RBD (yellow), FP (pink), HR1 (cyan). The remaining monomers are displayed in grey. **(C)** The structure of the spike protein with one RBD up and bound to ACE2 (PDB ID 7DF4) is shown to highlight the mutations used in this study located in the RBD in the context of its interaction with the receptor. The spike protein is represented in the same color scheme used in (B) and ACE2 is displayed in teal.

neutralization curves of the wild-type (WT, 614G) and original (614D) strains, as well as variants of concern B.1.1.7, B.1.351 and P.1 (containing all defining mutations, Fig 2A–2D and S1 Table) are shown in Fig 2E and the complete set of neutralization curves in S4 Fig. For each neutralization curve, we calculated the half-maximal neutralization titer ($NT_{50}$), defined as the reciprocal of the dilution at which infection was decreased by 50% (Fig 2F). Consistent with the literature, the ELISA titer was associated with the neutralizing titer of sera (S2 Table) [45,56].

In this paper, we used the Wilcoxon statistical test when comparing $NT_{50}$s to understand if variants are significantly different from the WT. However, to define whether variants are capable of evading neutralization, we use the 4-fold criteria, similar to what has been done with influenza virus, where differences in hemagglutination assays greater than 4-fold highlight the need for vaccine update [33,57,58]. Data show that, although the spike including all defining mutations of B.1.1.7 exhibited a modest reduction in neutralization titers relative to WT spike, it was still efficiently neutralized by convalescent sera, as observed before (Fig 2F and 2G) [32,59]. Conversely, the complete set of defining mutations of B.1.351 and P.1 consistently

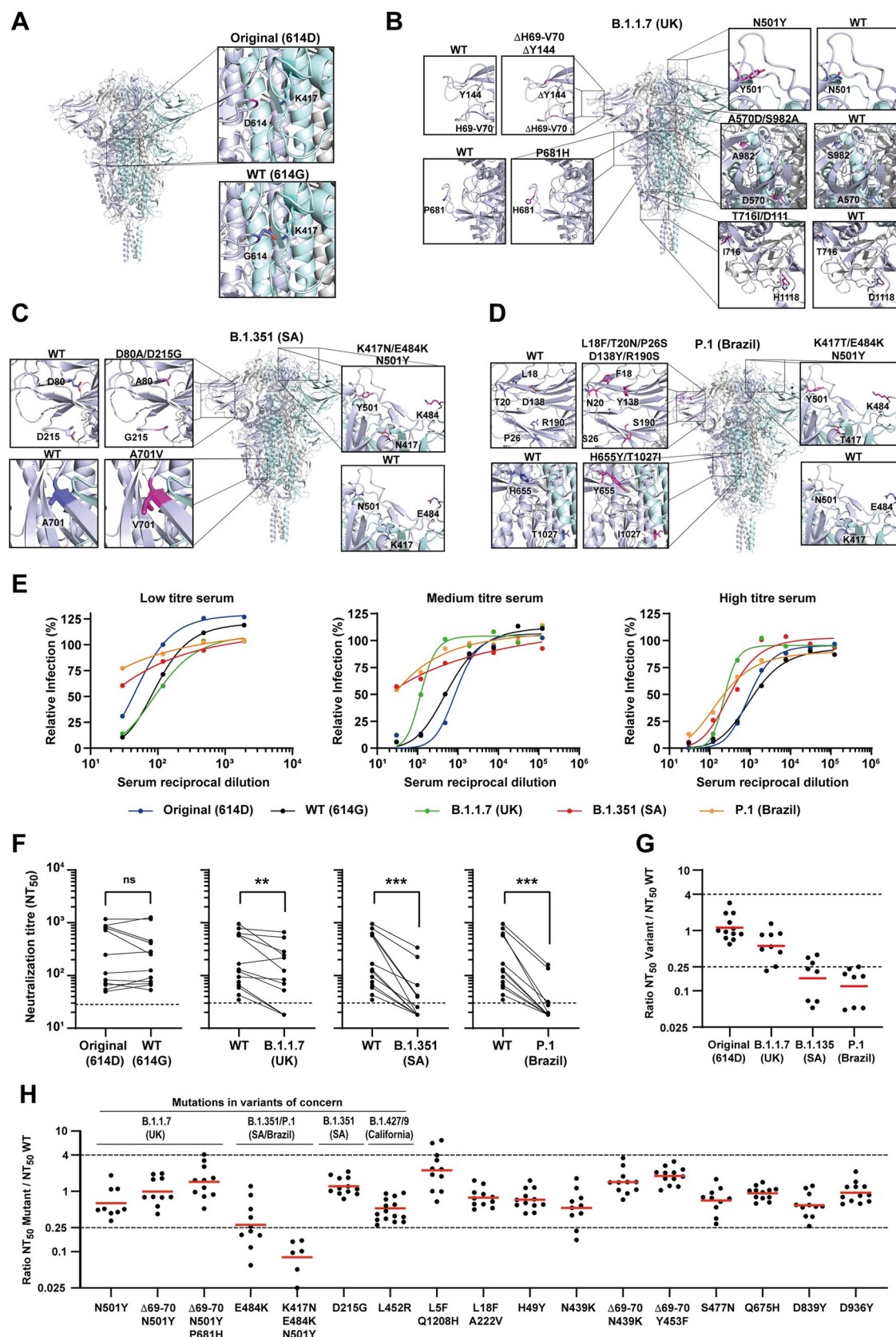

**Fig 2. Neutralization of SARS-CoV-2 spike mutants by convalescent human sera. (A-D)** SARS-CoV-2 spike secondary structure of the original strain (A), the B.1.1.7 variant first identified in the UK (B), the B.1.351 variant (SA) (C) and the P.1 variant (Brazil) (D). Insets show spike structure surrounding each mutated amino acid, and the corresponding structure in the WT. **(E-H)** Neutralization assays were performed by infecting 293T-ACE2 cells with pseudoviruses displaying WT or mutant SARS-CoV-2 spike, in the presence of serial dilutions of a convalescent human sera cohort (n = 12–16). **(E)** Representative neutralization curves of sera with low, medium, and high titers of anti-spike IgG antibodies against WT virus, original virus, and variants of concern B.1.1.7, B.1.351 and P.1. Relative infection is defined as the percentage of infection relative to cells infected in the absence of serum. **(F)** Paired analysis of neutralizing activity of each serum against WT vs original virus or variants of concern. $NT_{50}$ is defined as the inverse of the dilution that achieved a 50% reduction in infection. Dashed lines indicate the limit of detection of the assay ($NT_{50}$ = 30). ns, non-significant, **$p<0.01$, ***$p<0.001$ by two-tailed Wilcoxon matched-pairs signed-rank test. **(G)** Ratios of neutralization between variant and WT viruses. **(H)** Ratios of neutralization between mutant viruses with single, double, or triple mutations and WT virus. Red bars indicate the geometric mean. See S4 and S5 Figs and S2 Table for the complete set of neutralizing curves, neutralizing titers, anti-spike IgG titers determined by ELISA and the numerical values plotted in this figure.

showed a reduction in neutralization capacity higher than 4-fold and are considered significantly different from the WT virus (Fig 2G). We then analysed a series of single, double and triple mutations (Figs 2H, S4 and S5). We found that the mutations N501Y, Δ69-70/N501 and Δ69-70/N501+P681H, associated with B.1.1.7 variant, behave as WT spike. The mutation E484K alone reduced sera neutralization capacity by 3.6-fold [ratio $NT_{50}$ mutant/$NT_{50}$ WT of 0.028±0.37 (geometric mean ± SD)], and inclusion of additional mutations (K417N and N501Y) further decreased the neutralization (0.08±0.05), emphasizing the role of synergic mutations on immune evasion and underscoring the need for evaluating single and combined mutations on variants [48]. Other mutations outside the RBD did not significantly alter sera neutralization efficiency. However, we observed a reduction in neutralization in some sera against N439K mutant (0.53±0.45) as reported in [38], which was not observed when Δ69–70 was added (Figs 2H and S5), and also against L452R mutant (0.48±0.23) that is associated with the most recent variants of concern B.1.427/9, as reported in [60].

Importantly, we tested BNT162b2-elicited plasmas (collected after the first and second dosage of the Pfizer–BioNTech vaccine) for their capacity to neutralize the spike-pseudotyped particles expressing WT spike protein or mutants bearing all defining mutations of the variants of concern B.1.1.7, B.1.351 and P.1. Amongst 10 individuals, plasma collected 12 days after the administration of the first vaccine dose displayed anti-spike IgG titers from 1:450 to 1:12150 (S3 Table), and only one of them exhibited neutralizing capacity (S3 Table and S6 Fig). BNT162b2-elicited plasma collected 12 days after the administration of the 2nd dose had very high levels of anti-spike IgG (titers 1:36450–1:109350), higher than any sera collected upon natural infection, and all exhibited neutralizing activity against the variants of concern B. 1.351 and P.1, although with significantly lower efficiency than against the wild type or the B.1.1.7 variant (Fig 3).

Collectively, these data show that the spike-pseudotyped lentiviral particles used in this study, in which GFP is the output and is measured by high throughput and cheap methodologies, is suitable to quantitatively assess how single and synergetic mutations in full-length spike affect neutralizing-antibody responses.

## Frequency of contacts spike-antibodies and spike-ACE2

We then asked if we could find signatures in the spike protein leading to escape in antibody neutralization. To be able to escape antibodies while maintaining the ability to efficiently infect cells, the mutations introduced must destabilize the interaction with antibodies while keeping a high binding affinity for ACE2. Thus, we reasoned that mutations occurring in residues that are relevant for antibody binding but not very relevant for the interaction with ACE2 can be advantageous for the virus and, therefore, we needed to establish which amino acid residues are involved in antibody binding and in RBD binding.

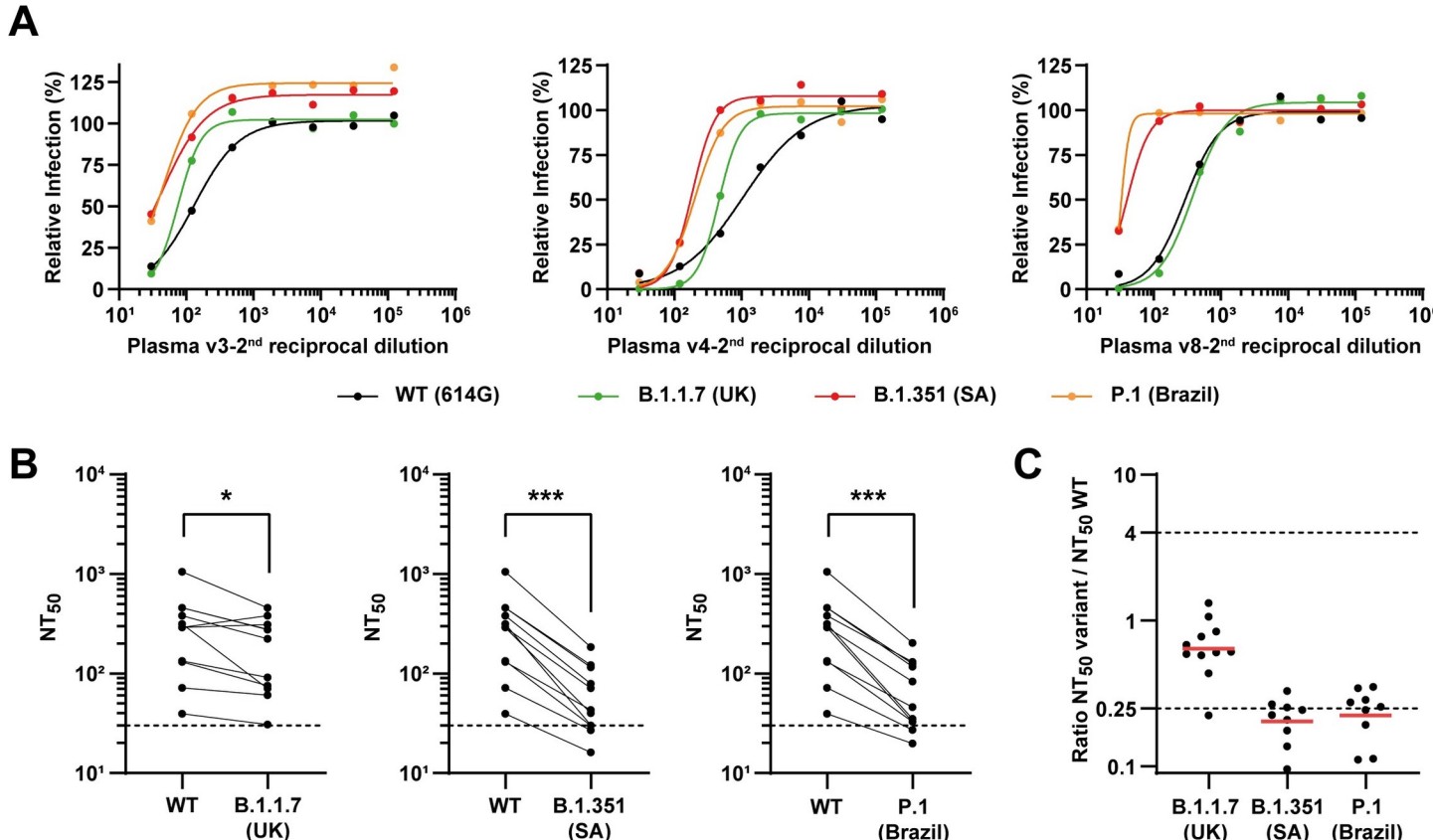

**Fig 3. Neutralization of SARS-CoV-2 variants of concern by post-vaccination plasma.** Neutralization assays were performed with pseudoviruses displaying WT SARS-CoV-2 spike or variants of concern B.1.1.7, B.1.351 and P.1, in the presence of serial dilutions of post-vaccination plasmas from 10 individuals, 12 days after the first and the second rounds of vaccination. **(A)** Representative neutralization curves of post-vaccination plasma against WT virus and variants of concern. **(B)** Paired analysis of neutralizing activity each of plasma against WT vs variants of concern. Dashed lines indicate the limit of detection of the assay ($NT_{50} = 30$). *$p<0.05$, ***$p<0.001$ by two-tailed Wilcoxon matched-pairs signed-rank test. **(C)** Ratios of neutralization between variant and WT viruses. Red bars indicate the geometric mean. See S6 Fig and S3 Table for the complete set of neutralizing curves, the anti-spike IgG titers of each plasma and the numerical values plotted in this figure.

To determine which spike protein residues are more frequently targeted by neutralizing antibodies, we analyzed 57 structures retrieved from the protein data bank (PBD) containing the spike protein (or only the RBD region) bound to antibodies (S4 Table). Our results revealed that the receptor binding motif (RBM), in RBD, is the region with the highest frequency of contacts (Fig 4A and 4B), consistent with what has been documented [11,61–64]. This region is important for the SARS-CoV-2 infection since it mediates contacts with ACE2 [50,65,66] and, for this reason, the binding of antibodies to this region would hinder ACE2 binding and, consequently, the entry of the virus in the cell.

In addition to contacts with the RBD, we also identified other regions of the spike protein with antibody binding sites. The antibodies FC05 and DH1050.1 bind to segments in the N-terminal region (between residues 143–152 and 246–257), region identified as an antigenic site [54], and 2G12 binds to residues 936–941 in the stalk (Fig 4B and 4C).

To choose the cut-off above which to select the most important residues for antibody binding, we built a histogram (S7 Fig). Three distinct groups appear: the first includes residues that have an interaction probability below 20%, the second includes residues whose interaction is between 20% and 45% and the last corresponds to frequent binders (interaction probability over 45%). Within this last group, we find Y449, L455, F456, E484, F486, N487, Y489, Q493,

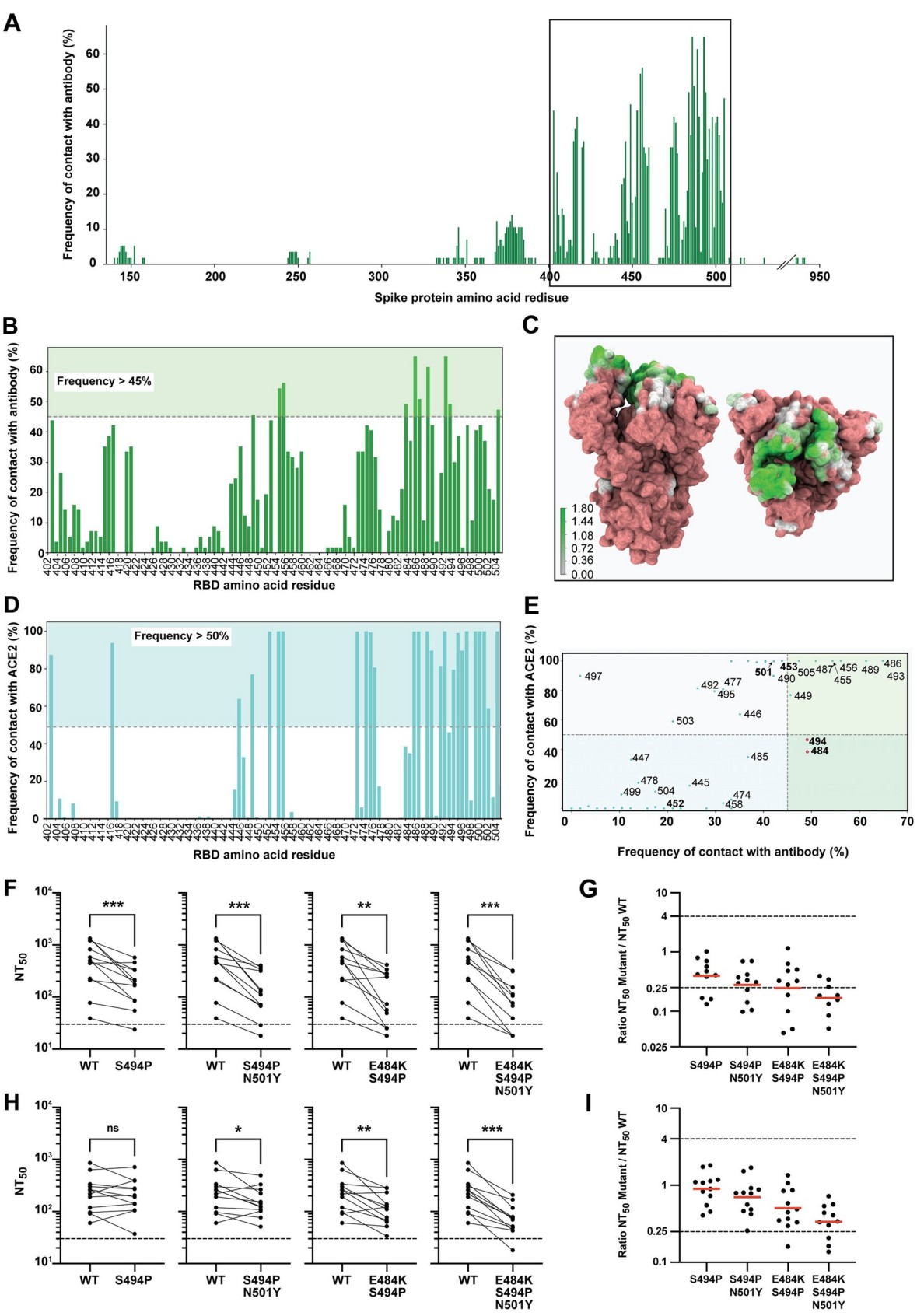

**Fig 4. Spike protein-antibody and RBD-ACE2 contact analysis. (A)** Histogram of the contact frequency of each spike residue with antibodies using a cut-off value 4.5 Å to define a contact. **(B)** Zoom in of the histogram shown in (A) in the region comprising the residues with the highest frequency of antibody contacts (located in the RBD). **(C)** Antibody contact map represented on the spike protein surface, seen from front (left side) and from top (right side); the map was built with VMD [108] and the color gradient is used to represent the log of the frequency of contact with Ab as shown in the scale bar. **(D)** Histogram of the contact frequency of RBD residues with ACE2, obtained from MD simulations, using a cut-off value 4.5 Å to define a contact. **(E)** Scatter plot correlating the RBD residues' frequency of contact with ACE2 and antibodies; each point represents one residue and those that were studied experimentally are highlighted in bold; the quadrants are filled with different colors to highlight the relevance of different residues as potential mutation hotspots, with the most relevant residues located in quadrant IV and shown as red points. **(F-I)** Neutralization assays were performed against pseudoviruses displaying WT or S494P spike mutants, in the presence of serial dilutions of convalescent sera (F, G) or post-vaccination sera 1 month after the first and the second rounds of vaccination (H, I). **(F, H)** Paired analysis of neutralizing activity of convalescent sera (F) or post-vaccination sera (H) against WT vs S494P, S494P/N501Y, E484K/S494P or E484K/S494P/N501Y mutants. Dashed lines indicate the limit of detection of the assay ($NT_{50}$ = 30). ns, non-significant, *p<0.05, **p<0.01, ***p<0.001 by two-tailed Wilcoxon matched-pairs signed-rank test. **(G, I)** Ratios of sera neutralization between S494P mutants and WT viruses. Red bars indicate the geometric mean. See S7 and S8 Figs and S4 Table for protein-antibody contact frequency histogram, overall representation of amino acid residues relevant for antibody and ACE2 binding and list of references of the structures of the antibodies used, respectively; S9 and S10 Figs and S5 and S6 Tables for the complete set of neutralizing curves, the anti-spike IgG titers of each serum and the numerical values plotted in this figure.

S494, and Y505 residues (Fig 4A and 4B). More than half of these residues have hydrophobic side chains, showing that, within the batch of antibodies we studied, the most common binding mode is through hydrophobic contacts. These residues are important for antibody binding, which means that mutations within this group may enable the virus to escape antibodies from patients previously exposed to the WT or another variant, or people vaccinated using WT sequences of spike. In fact, prior studies have shown that mutations on most of these residues influence antibody escape [2,67]. For residues L455, F456 and F486, mutations for less hydrophobic residues have been shown to reduce binding by polyclonal antibodies [67], and mutants at site 487 escape human monoclonal antibodies COV2-2165 and COV2-2832 [2,63]. Interestingly, our results also show that there is a high prevalence of antibodies binding to E484 (~50%, Fig 4B).

To determine which amino acid residues in spike engage in ACE2 binding, we performed molecular dynamics (MD) simulations of the RBD bound to ACE2. These allowed us to determine which RBD residues are relevant for binding to ACE2, and the persistence of these interactions (Fig 4D). Combining the information on antibody and ACE2 binding allows us to predict which are the most relevant amino acid residues (high affinity for antibodies and low affinity for ACE2). Using these criteria, two residues stand out: E484 and S494 (Fig 4E, IV quadrant circled in red). These results are consistent with the evidence that E484 is an important mutation site and bring to light a new relevant site: S494. Mutations in S494 have been found in circulating strains, and the S494P mutation was shown to reduce the binding by polyclonal plasma antibodies [67] whilst having no [4] or modest effect [68] in RBD-ACE2 binding.

To elucidate the role of this mutation in antibody neutralization, we used our neutralization assay and found that mutation S494P alone leads to a reduction in neutralization by convalescence sera that is significant (Figs 4F and S9, and S5 Table) despite not reaching our 4-fold criteria (ratio $NT_{50}$ of mutant in relation to WT of 0.39±0.28, Fig 4G). This value was further reduced when this mutation was combined with N501Y (0.28±0.21, Fig 4F and 4G), or when combined with E484K (0.25±0.32, Fig 4F and 4G). We constructed these spike mutants because we aimed at understanding if mutations that increased the ability to bind ACE2, such as N501Y, could synergize with mutations escaping antibody neutralization. Interestingly, the E484K/S494P/N501Y triple mutant reduced neutralization over the 4-fold criteria applied (0.17±0.34, Fig 4F and 4G). We also tested the ability of these mutants to evade neutralization by post-immunization sera, but the effect was less pronounced (Figs 4H, 4I and S10, and S6 Table). In fact, S494P alone did not significantly reduce neutralization, becoming significant

when N501Y (0.70±0.4), or E484K (0.5±0.3) mutations were introduced. The triple mutant displayed the more significant reduction in neutralization, despite not reaching the 4-fold difference (0.34±0.18, Fig 4H and 4I). Importantly, S494P mutation appeared independently on several occasions and is under positive selection [68], and is now a variant in monitoring, especially when combined with E484K or N501Y [20,69]. Our results indicate that the S494P mutation can facilitate the escape of the virus from antibodies, maintaining the ability to bind to the receptor and enter host cells, and should be surveilled worldwide.

Some residues that are important for antibody binding are also involved in frequent interactions with ACE2. Seven of these residues remained bound to ACE2 throughout the whole simulation in all the replicates: L455, F456, F486, N487, Y489, Q493 and Y505 (Fig 4D, top residues in quadrant II). This group includes residue F456, which was found to be a hotspot for escape mutations [67]. Although this may seem contradictory, the same study shows that mutations in this residue were very rarely found in nature and, because of lack of prevalence, were not analyzed [67]. We hypothesize that the low mutation rate for this site may be due to the high relevance of this residue for RBD-ACE2 binding. In fact, deep mutational analysis shows that when mutating F546 to any amino acid residue besides its SARS-CoV-1 counterpart (a leucine), the ACE2 binding affinity is reduced [4]. Even though a viral variant with a single mutation at the site 456 could escape antibodies, it would be unfit to subsist in nature, due to a reduced ability to infect host cells. In addition, some residues such as N501, S417, L452 were shown to be strong ACE2, but weak antibody binders, raising the possibility that these amino acids may undergo mutations that cooperatively synergize with antibody binders to increase the viral fitness of immune escape competent SARS-CoV-2 variants. Interestingly, N501Y, L452R and S417N have been associated with increased viral transmission [18,19,26,36] and, when combined with E484K, augmented resistance to neutralizing antibodies.

Together, the analysis presented here is supported by experimental evidence on mutation-driven binding to ACE2 and escape to antibodies [2–4], as well was with the emergence of natural variants [28,32,38,55,70–72], and may be used to predict mutation hotspots.

## Discussion

It is critical to understand SARS-CoV-2 signatures that allow escaping neutralizing antibodies, which could pose challenges to vaccination and render therapies ineffective.

We developed spike-pseudotyped lentiviral particles for high-throughput quantitation that express GFP upon entering cells, contributing to the toolkit of neutralizing assay methodologies [73–75]. Our method of reading fluorescent signals in 96 wells plates using the GloMax Explorer System is suitable to assess the neutralization activity of sera/plasma from individuals infected naturally or vaccinated, and to screen for antiviral drugs that block viral entry, such as therapeutic antibodies, in biosafety level 2 settings, which greatly facilitates the procedure and broadens its usage. It can be easily adapted to include single and multiple mutations, as observed in the present work for sera neutralizing activity (Fig 2). Our results with BNT162b2-elicited plasma (Fig 3) agree with previous publications showing resistance to neutralization by B.1.351 and P.1 lineages [28,29,33,47,76] and validate our neutralization assay. The fact that only mutations containing the E484K substitution promote immune escape, although this effect increases with synergic mutations that improve viral binding to ACE2 (K417N and N501Y), may be a consequence of lack in immunological selective pressure, a recognized driver of evolution [58,77–79], as a large proportion of the population remains susceptible to SARS-CoV-2 infection. However, the emergence of this escape mutant suggests that the continued circulation of the virus may, in the future, impose further immunological constraints and result in viral evolution, as seen for influenza A virus [58,80]. How changes in

SARS-CoV-2 will impact the circulation of the virus is not known, and the future will also elucidate whether vaccination will shape evolution of SARS-CoV-2 and permit reinfections. At the moment, reinfections are considered rare events [81,82], but reports that SARS-CoV-2 escape mutants were shown to drive resurgence of cases upon natural infection [23,83] are indicative that viral dynamics in the population may change [26,84]. Therefore, the development of methods able to predict hotspots of immune evasion are in demand. The surveillance of circulating strains across time, the evaluation of the type and duration of host immune responses upon natural infection and vaccination, and the identification of antibodies that efficiently control SARS-CoV-2 are essential measures to understand and control SARS-CoV-2 infection and host response [85–87]. We argue that the structural information on complexes between antibodies, or ACE2, and spike variants may be integrated with mutational maps and their phenotypic characterization [2–4] to predict hotspots of immune evasion.

In this work, we analyzed by neutralization assays how a full-length spike carrying single or multiple mutations dispersed throughout the protein affected the conformation of the RBD. With this assay, we probed how antibodies from infected patients or vaccinated individuals block viral entry. We used structural information on the protein complexes spike-antibodies and spike-ACE2 to define the non-overlapping frequency of interactions using a cut-off value of 4.5 Å distance to define a contact. Given that only a few amino acid residues are involved in protein-protein interfaces [88], including in antibody-antigen recognition, we aimed at identifying which amino acid residues may evolve antibody escape mutants without affecting viral entry. With this approach, we identified two amino acid residues at the RBD—positions 484 and 494 (Fig 4E)—that frequently engage in interactions with antibodies but not with ACE2. We observed that E484 is relevant for antibody binding (Fig 2), in agreement with our and previous experimental results, and with the occurrence of this mutation in two variants that are becoming highly prevalent and were shown drive resurgence of infection in sites with high levels of seroconversion [2,33,57,71,72,83]. Additionally, we found that mutation of residue 494 may be problematic, either alone or combined with synergetic mutations, as it reduces neutralization competency of convalescent sera, and thus facilitates antibody escape without substantially altering the affinity for the ACE2 binding (Figs 4E, 4G and S8). The mutation S494P has been found in nature with a prevalence of 0.81% of all sequenced genomes (March 2021, S1 Fig) and has already been observed in combination with mutation E484K (reported on the 22nd of October 2020). Of note, current prevalence of B.1.351 and of P.1 is 1.9% and 0.47%, respectively, being in the same range as S494P. We posit that the prevalence of mutations in amino acids 484, 494 and 501 will increase with viral circulation and/or spike seroconversion and should be surveilled worldwide. We observed a similar pattern relative to the substitution L452R/Q in India, in which the E484Q mutation was acquired posteriorly. At the moment, however, with the majority of the population still susceptible to viral infection, mutations that increase viral transmission, such as N501Y [84] have a selective advantage. In the future, mutations escaping host antibodies whilst increasing binding to ACE2 may become prevent.

Our approach has some caveats. Regarding the neutralization assays, and despite spike being the immunodominant protein targeted by antibodies [7–11], other viral proteins may contribute, even if in a small proportion, to neutralization activity *in vivo*, which we are unable to detect using spike-pseudotyped particles. In addition, the sera/plasma we used were obtained at a fixed time interval. Future experiments should repeat the analysis using sera/plasma of people infected with known genotypes, or from different geographical areas and obtained at different times along the epidemic dynamics. Finally, we inspected a handful of mutations and not a comprehensive mutational map across spike [2,4]. However, we traced several mutations across spike in single and multiple combinations, and thus, we conclude that our analysis is relevant to inform on mutation-driven escape viruses to antibodies. In

addition, we only investigated antibody-neutralization responses but other mutations in spike may contribute to overall antibody escape, and hence regulate the host response to infection, which we are unable to detect by our assay.

The analysis of frequency of interactions between spike-antibodies or ACE2 has some issues associated as well. The pool of complexes spike protein/RBD and antibodies analyzed here is not necessarily representative of the universe of antibodies found in COVID-19 patients, since it is limited by the availability of structural information and is biased to monoclonal antibodies (S4 Table). All the antibodies analyzed are neutralizing antibodies, which is not the case in nature [54]. Additionally, many structural studies have focused on the interaction with the RBD, while other regions have not been so exhaustively analyzed, which creates a bias in terms of the number of RBD binding antibodies with known structure [50,65,66]. Limitations of the methods used, such as crystallographic contacts, can also influence the results. Nevertheless, its combination with data from the molecular simulation and from experiments helps to predict and explain the emergence of new mutations.

In conclusion, our analysis of the spike protein-antibody contacts revealed that (within the available data set) the receptor binding motif, in the receptor binding domain, is an important region for antibody binding, which makes sense, since antibody binding to this region can prevent the interaction with ACE2 and ultimately impair viral infection. Our results are in very good agreement with experimental data on mutation-driven escape to polyclonal antibodies and with the emergence of natural variants, which were reported to re-circulate in previously exposed people [83]. Thus, this analysis represents a simple strategy to predict mutation hotspots that should be kept under surveillance. The predictions will become even better once the pool and diversity of available structures increases.

## Materials and methods

### Ethics statement

This study was approved by the Ethics committees of Centro Hospitalar Lisboa Ocidental and Centro Hospitalar Lisboa Central, in compliance with the Declaration of Helsinki, and follows international and national guidelines for health data protection. All participants provided informed written consent to take part in the study.

### Global variants frequencies

A variant surveillance report was retrieved from GISAID on March 26th, 2021. A total of 864131 submissions were used to establish the world frequency of each variant using specific amino acid mutations (e.g., S_D614G, etc.) by using command line search text tools. Counts were then divided by week and year, using available metadata for collection date and just variants over indicated prevalence are reported and plotted per graph.

### Cells and plasmids

Human hepatocellular carcinoma HuH-7 cells (a kind gift from Dr Colin Adrain, Instituto Gulbenkian de Ciência, Portugal), Human Embryonic Kidney 293T (provided by Prof Paul Digard, Roslin Institute, UK) and 293ET cells (from Dr Colin Adrain) were maintained in Dulbecco's Modified Eagle Medium (DMEM, Gibco, 21969035) supplemented with 10% fetal bovine serum (FBS, Gibco, 10500064), 1% penicillin/streptomycin solution (Biowest, L0022) and 2mM L-glutamine (ThermoFisher, 25030024), at 37˚C and 5% $CO_2$ atmosphere. Plasmids used in this study were obtained as follows: pLEX.MSC was purchased from Thermo Scientific; psPAX2 and pVSV.G were a kind gift from Dr Luís Moita (Instituto Gulbenkian de Ciência,

Portugal); pEGFP-N1 was provided by Dr Colin Crump (University of Cambridge, UK); and pCAGGS containing the SARS-Related Coronavirus 2, Wuhan-Hu-1 Spike Glycoprotein Gene (pCAGGS-SARS-CoV-2-S, NR-5231) was obtained through BEI Resources.

## Cloning and spike protein mutations

To facilitate incorporation of spike protein into lentiviral pseudovirions, the C-terminal 19 amino acids encompassing the ER-retention domain of the cytoplasmic tail of spike were deleted. This construct, named SARS-CoV-2 S$_{trunc}$, was generated by PCR amplification of the C-terminal region of S without amino acids 1255–1273 (a premature STOP codon was added in the reverse primer), using pCAGGS-SARS-CoV-2-S as template (primers in S7 Table). The amplified sequence was cloned back into pCAGGS-SARS-CoV-2-S, between *Age*I and *Sac*I restriction sites, generating the expression vector pCAGGS-SARS-CoV-2-S$_{trunc}$. Mutations were introduced in the spike protein by site-directed mutagenesis, with the Quickchange Multi Site-Directed Mutagenesis kit (Agilent) following the manufacturer's instructions (primers in S7 Table). Lentiviral reporter plasmid pLEX-GFP was produced by PCR amplifying GFP from pEGFP-N1 (primers in S7 Table) and cloning the insert into *BamH*I–*Xho*I restriction sites in the multi-cloning site of pLEX.MCS vector. For lentiviral plasmid pLEX-ACE2 production, human ACE2 coding sequence was amplified from Huh7 cDNA and cloned into pLEX. MCS, using *Xho*I and *Mlu*I restriction sites (primers in S7 Table).

## Production of 293T cells stably expressing human ACE2 receptor

To produce VSV-G pseudotyped lentiviruses encoding the human ACE2, 293ET cells were transfected with pVSV-G, psPAX2 and pLEX-ACE2 using jetPRIME (Polyplus), according to manufacturer's instructions. Lentiviral particles in the supernatant were collected after 3 days and were used to transduce 293T cells. Three days after transduction, puromycin (Merck, 540411) was added to the medium, to a final concentration of 2.5μg/ml, to select for infected cells. Puromycin selection was maintained until all cells in the control plate died and then reduced to half. The 293T-ACE2 cell line was passaged six times before use and kept in culture medium supplemented with 1.25 μg/ml puromycin. ACE2 expression was evaluated by flow cytometry (S2 Fig).

## Production and titration of spike pseudotyped lentiviral particles

To generate spike pseudotyped lentiviral particles, 3x10$^6$ 293ET cells were co-transfected with 8.89μg pLex-GFP reporter, 6.67μg psPAX2, and 4.44μg pCAGGS-SARS-CoV-2-S WT or mutants (or pVSV.G, as a control), using jetPRIME according to manufacturer's instructions. The virus-containing supernatant was collected after 3 days, concentrated 10 to 20-fold using Lenti-X™ Concentrator (Takara, 631231), aliquoted and stored at -80˚C. Pseudovirus stocks were titrated by serial dilution and transduction of 293T-ACE2 cells. At 24h post transduction, the percentage of GFP positive cells was determined by flow cytometry, and the number of transduction units per mL was calculated.

## Flow cytometry

Flow cytometry was performed as in [89]. In brief, HEK 293T and 293T-ACE2 cells were prepared for flow cytometry analysis by detaching from the wells with trypsin, followed by fixation with 4% formaldehyde). For analysis of ACE2 expression, cells were stained with a primary antibody against ACE2 (4μg/ml, R&D Systems, catalog no. AF933) followed by a secondary antibody labelled with Alexa Fluor 568 (1:1000; Life Technologies). Analysis of cell

populations was performed in a Becton Dickinson (BD) LSR Fortessa X-20 equipped with FACS Diva and FlowJo (BD, Franklin Lakes, NJ) software's.

## Human convalescent sera and post-vaccination plasma/serum

Venous blood was collected by standard phlebotomy from health care providers who contracted COVID-19 during spring/summer 2020, as tested by RT-PCR on nasopharyngeal swabs. Serum was prepared using standard methodology and stored at -20˚C. During this period, the variant B.1.1, encoding the WT spike protein, was prevalent in Portugal.

Peripheral blood from health care workers was collected by venipuncture into EDTA tubes at day 12 and day 33 post-immunization with Pfizer BTN162b2 mRNA vaccine and immediately processed. Alternatively, post-vaccination serum was collected at day 22 and day 51 post-immunization with Pfizer BTN162b2 mRNA vaccine. Peripheral blood was diluted in PBS 1x (VWR), layered on top of biocoll (Biowest) and centrifuged at 1200g for 30 min without break. Plasma was collected to cryotubes and stored at -80˚C ultra-low freezer until subsequent analysis.

## ELISA assay

Direct ELISA was used to quantify IgG anti-full-length Spike in convalescent sera. The antigen was produced as described in [90]. The assay was adapted from [91] and semi-automatized to measure IgG in a 384-well format, according to a protocol to be detailed elsewhere. For titer estimation, sera were serially diluted 3-fold starting in a 1:50 dilution and cut off was defined by pre-pandemic sera (mean + 2 standard deviation).

ELISA assay on post-vaccination plasma and serum was performed based on the protocol [92] and modified as described in Gonçalves *et al.* [93]. Briefly, 96 well plates (Nunc) were coated with 50 μl of trimeric spike protein at 0.5 μg/mL and incubated overnight at 4˚C. On the following day, plates were washed three times with 0.1% PBS/Tween20 (PBST) using an automatic plate washer (ThermoScientific). Plates were blocked with 3% bovine serum albumin (BSA) diluted in 0.05% PBS/T and incubated 1h at room temperature. Samples were diluted using 3-fold dilution series starting at 1:50 and ending at 1:10,9350 in 1% BSA-PBST/T and incubated 1h at room temperature. Plates were washed three times as previously and goat anti-human IgG-HRP secondary antibodies (Abcam, ab97215) were added at 1:25,000 and incubated 30 min at room temperature. Plates were washed three times and incubated ~7min with 50 μl of TMB substrate (BioLegend). The reaction was stopped with 25μl of 1M phosphoric acid (Sigma) and read at 450nm on a plate reader (BioTek). Each plate contained 6 calibrator samples from two high-, two medium-, and two low- antibody producer from adult individuals collected at Hospital Fernando Fonseca that were confirmed positive for SARS-CoV-2 by RT-PCR from nasopharyngeal and/or oropharyngeal swabs in a laboratory certified by the Portuguese National Health Authorities [93]. As negative control, we used pre-pandemic plasma samples obtained from healthy donors collected prior July 2019. The endpoint titer was defined as the last dilution before the absorbance dropped below $OD_{450}$ of 0.15.

## Neutralization assay

Heat-inactivated serum and plasma samples were four-fold serially diluted over 7 dilutions, beginning with a 1:30 initial dilution. Dilutions were then incubated with spike pseudotyped lentiviral particles for 1h at 37˚C. The mix was added to a pre-seeded 96 well plate of 293T-ACE2 cells, with a final MOI of 0.2. At 48h post-transduction, the fluorescent signal was measured using the GloMax Explorer System (Promega). The relative fluorescence units were normalized to those derived from the virus control wells (cells infected in the absence of

plasma or serum), after subtraction of the background in the control groups with cells only. The half-maximal neutralization titer ($NT_{50}$), defined as the reciprocal of the dilution at which infection was decreased by 50%, was determined using four-parameter nonlinear regression (least squares regression without weighting; constraints: bottom = 0) (GraphPad Prism 9). To assess the specificity of our assay, an anti-spike antibody neutralization assay and an RBD competition assay were performed in SARS-CoV-2 spike pseudotyped and vesicular stomatitis virus (VSV) G pseudotyped lentiviral particles, in parallel (S3 Fig). The anti-spike antibody neutralization assay was performed as above, starting with a concentration of 50μg/ml (Thermo Fisher Scientific, PA5-81795) and three-fold serially diluting over 7 dilutions. For the RBD competition assay, 293T-ACE2 cells were pre-incubated with three-fold serial dilutions of SARS-CoV-2 spike's receptor binding domain, starting with a concentration of 400μg/ml. After 1h incubation at 37˚C, supernatant was aspirated and pseudoviruses were added at an MOI of 0.2. The half maximal inhibitory concentration ($IC_{50}$) was determined using four-parameter nonlinear regression (least squares regression without weighting; constraints: bottom = 0) (GraphPad Prism 9).

## Spike protein-antibody contact analysis

To determine which residues of the SARS-CoV-2 spike protein contribute the most for the binding of antibodies, we studied 57 PDB structures containing the spike protein (or only the RBD region) bound to antibodies (S4 Table). These complexes were chosen based on their availability in the PDB repositorium [94] and are all neutralizing antibodies. We first used the MDAnalysis library [95,96] to identify the residues of the antibody and the spike protein located in the interface between the two proteins. This was done for all PDB structures using in-house Python scripts. To determine the relevance of each spike protein/RBD residue in the binding, a distance cut-off value of 4.5 Å was applied as a criterion (i.e., a contact is observed when two residues have a minimum distance lower than 4.5 Å). Finally, the numerical python library (NumPy [97]) was used to calculate the frequency of contact for each spike protein residue with an antibody residue, from the sample of PDB structures analyzed.

## MD simulation of the RBD-ACE2 complex

Molecular dynamics (MD) simulations of the RBD bound to the ACE2 protein were performed with the GROMACS 2020.3 package [98], using the Amber14sb [99] force field, starting from the 6m0j structure [49], in a truncated dodecahedron box filled with water molecules (minimum of 1.2 nm between protein and box walls). The TIP3P water model [100] was used and the total charge of the system (-23, including the constitutive $Zn^-$ and $Cl^-$ ions bound to ACE2) was neutralized with 23 $Na^+$ ions. Additional $Na^+$ and $Cl^+$ ions were added to the solution to reach an ionic strength of 0.1M. The system was energy-minimized using the steepest descent method for a maximum of 50000 steps using position restraints on the heteroatom positions by restraining them to the crystallographic coordinates using a force constant of 1000 kJ/mol in the X, Y and Z positions. Before performing the production runs, an initialization process was carried out in 5 stages of 100 ps each. Initially, all heavy-atoms were restrained using a force constant of 1000 kJ/mol/nm, and at the final stage only the only C-α atoms were position-restrained using the same force constant. In the first stage, the Berendsen temperature algorithm [101] was used to initialize and maintain the simulation at 300 K, using a temperature coupling constant of 0.01 ps, without pressure control. The second stage continues to use the Berendsen temperature algorithm but with a force constant of 0.1 ps. The third stage kept the same temperature control settings, but introduced pressure coupling with the Berendsen pressure algorithm [101] with a pressure coupling constant of 5.0 ps and applied isotropically.

The fourth stage changed the temperature algorithm to V-rescale [102], with a temperature coupling constant of 0.1 ps, and the pressure algorithm to Parrinello-Rahman [103] with a pressure coupling constant of 5.0 ps. The fifth stage is equal to the fourth stage, but position restraints are only applied to C-α atoms. During the simulation, the equations of motion were integrated using a timestep of 2 fs. The temperature was maintained at 300 K, using the V-rescale [102] algorithm with a time constant of 0.1 ps, and the pressure was maintained at 1 bar using the Parrinello–Rahman [103] pressure coupling algorithm, with a time constant of 5 ps; pressure coupling was applied isotropically. Long-range electrostatic interactions were treated with the PME [104,105] method, using a grid spacing of 0.12 nm, with a cubic interpolation. The neighbor list was updated every twenty steps and the cutoff scheme used was Verlet with 0.8nm as the real space cut-off radius. All bonds were constrained using the LINCS algorithm [106]. The system was simulated for 8 μs in 5 replicates (to a total of 40 μs).

### RBD-ACE2 contacts frequency

To determine the residues that contribute the most for the interaction between RBD and ACE2, and the persistence of these interactions, we performed a contact analysis throughout the simulation. We started by eliminating the first equilibration μs of all replicates. The MDA-analysis library [95,96] was then used to pinpoint the residues of the RBD that are in contact with the ACE2 protein. A distance cut-off value of 4.5 Å was applied as a criterion (i.e., a contact is observed when two residues have a minimum distance lower than 4.5 Å). Finally, we determined the percentage of time for which a given RBD residue is at less than 4.5 Å of ACE2.

### Homology-based models of spike protein variants

Homology-based models of all the variants analyzed in this study were generated using the software Modeller [107], version 9.23, using the structure of the WT enzyme (PDB code: 6XR8) [12] as a template. The protocol used only optimizes the atoms belonging to the mutated residues and the residues that are located within a 5 Å radius from these residues, maintaining the remaining atoms fixed with the coordinates found in the template structure. The optimization parameters were set to the software default values. The final model corresponds to the one with the lowest value of the DOPE function out of 20 generated structures.

## Supporting information

**S1 Fig. Temporal worldwide prevalence of spike mutations considered in this study from the 29th of December 2019 until the 25th of March 2021.** To investigate the global frequency of SARS-CoV-2, the variant surveillance dataset was retrieved on March 25th, 2021, from GISAID (Shu Y, McCauley J., 2017). From this dataset and using text mining tools, the number and frequency of predominant variants was determined from spike protein mutations and displayed per month according to variant collection date. The mutations are divided onto 3 graphs in which frequencies are up to 100% (1st graph), 8.5% (2nd graph) and 1.75% (3rd graph) for easier visualization. The frequencies are not cumulative but individual representations of each mutation.
(TIF)

**S2 Fig. Histogram showing expression of human ACE2 by 293T-ACE2 cell line (blue) compared to parental 293T cells (green).** ACE2 expression was measured by flow cytometry, after staining with a goat anti-ACE2 antibody (R&D Systems) followed by staining with an anti-

goat antibody conjugated to Alexa 568 (Life Technologies).
(TIF)

**S3 Fig. Specificity of the *in vitro* neutralization assay with SARS-CoV-2 spike pseudotyped lentiviral particles. (A)** SARS-CoV-2 spike specific antibody was tested for neutralization activity against SARS-CoV-2 spike and vesicular stomatitis virus (VSV) G pseudotyped lentivirus. **(B)** Neutralization assay of SARS-CoV-2 spike and VSV G pseudoviruses in the presence of spike's receptor binding domain (RBD). 293T-ACE2 cells were pre-incubated with serial dilutions of RBD before infection with each virus. Half maximal inhibitory concentration ($IC_{50}$) was calculated for both assays, when possible.
(TIF)

**S4 Fig. Neutralization curves of spike mutants by human sera. Related to Figs 2 and S5 and S2 Table.** Sera from 16–20 individuals were tested for neutralization of WT and 22 viruses. Sera were classified into 4 categories: Negative, Low anti-spike IgG titer (≤1:150), Medium titer (1:450) and High titer (≥1:1350). Triplicates were performed for each tested serum dilution. Error bars represent standard deviation.
(TIF)

**S5 Fig. Neutralization titers of human sera against spike mutants. Related to Figs 2 and S4 and S2 Table.** Paired analysis of neutralizing activity of each convalescent serum against WT vs mutant virus. $NT_{50}$ is defined as the inverse of the dilution that achieved a 50% reduction in infection. Dashed lines indicate the limit of detection of the assay ($NT_{50} = 30$). ns, non-significant, *$p<0.05$, **$p<0.01$, ***$p<0.001$ by two-tailed Wilcoxon matched-pairs signed-rank test.
(TIF)

**S6 Fig. Neutralization curves of post-vaccination plasma against spike mutants. Related to Fig 3 and S3 Table.** Plasma was collected from 10 individuals 12 days after the first and the second rounds of vaccination and was tested for neutralization of WT virus and variants of concern. Triplicates were performed for each tested plasma dilution. Error bars represent standard deviation.
(TIF)

**S7 Fig. Spike protein-antibody contact frequency histogram.** This was computed by dividing the frequency of interaction of spike protein residues with antibodies in 100 bins and calculating the fraction of residues that fall into each bin.
(TIF)

**S8 Fig. Relevant residues for antibody and ACE2 binding. (A)** Crystal structure of the SARS-CoV-2 RBD complexed with ACE2. ACE2 is shown as a purple surface and the RBD is shown in a cyan cartoon representation with key antibody interacting residues depicted as sticks. **(B)** Zoom in to the RBM region of the RBD. Residues relevant for antibody binding (>35% frequency of contact) are depicted as sticks. Of these, the ones with an antibody binding probability higher than 45% have a green label, and those that also have a low frequency of binding to ACE2 (<50%) are labelled in pink.
(TIF)

**S9 Fig. Neutralization curves of WT and S494P spike mutants by human sera. Related to Fig 4 and S5 Table.** Sera from 19 individuals were tested for neutralization of WT and S494P mutant viruses. Sera were classified into 4 categories: Negative, Low anti-spike IgG titer (≤1:150), Medium titer (1:450) and High titer (≥1:1350). Triplicates were performed for each

tested serum dilution. Error bars represent standard deviation.
(TIF)

**S10 Fig. Neutralization curves of post-vaccination serum against spike mutants. Related to Fig 4 and S6 Table.** Serum was collected from 8 individuals 1 month after the first and the second rounds of vaccination, and was tested for neutralization of WT virus and S494P mutants. Triplicates were performed for each tested serum dilution. Error bars represent standard deviation.
(TIF)

**S1 Table. Details of the mutants created and used in this study.** Values of current frequency are relative to available sequences of week 11 of 2021 (with a bias corresponding to the efforts that countries/regions deploy to SARS-CoV-2 genome surveillance).
(DOCX)

**S2 Table. IgG antibody titers against SARS-CoV-2 spike protein and neutralizing titers (NT$_{50}$) of convalescent sera against WT and mutant spike pseudoviruses.**
(DOCX)

**S3 Table. IgG antibody titers against SARS-CoV-2 spike protein and neutralizing titers (NT$_{50}$) against WT and variant pseudoviruses of plasma from vaccinated individuals, collected 12 days after the first and after the second doses of the vaccine.**
(DOCX)

**S4 Table. Summary of antibodies studied in the spike-antibody complexes.** These antibodies were chosen due to the availability of their structure resolved together with the spike (or just the RBD region) in the PDB repository [94].
(DOCX)

**S5 Table. IgG antibody titers against SARS-CoV-2 spike protein and neutralizing titers (NT$_{50}$) of convalescent sera against WT and mutant spike pseudoviruses.**
(DOCX)

**S6 Table. IgG antibody titers against SARS-CoV-2 spike protein and neutralizing titers (NT$_{50}$) against WT and mutant pseudoviruses of serum from vaccinated individuals, collected 1 month after the first and after the second doses of the vaccine.**
(DOCX)

**S7 Table. Primers used in this study.**
(DOCX)

## Acknowledgments

The authors acknowledge Sean Whelan (Washington University School of Medicine St Louis) for providing reagents, João F. Viana and his team for organizing sera collection at Centro Hospitalar Lisboa Ocidental in Portugal, and the IGC COVID-19 team (Patrícia C. Borges, Christian Diwo, Paula Matoso, Vanessa Malheiro, Lígia A. Gonçalves, Nádia Duarte, Ana Brennand, Lindsay Kosack and Onome Akpogheneta) for processing and ranking samples, for this paper and during early settings. We are grateful to the Unit of Genomics of the Instituto Gulbenkian de Ciência for technical support, sample processing and data collection. The authors thank Isabel Gordo and Mónica Dias (Instituto Gulbenkian de Ciência) for helpful discussions and Rute Castro and Paula M. Alves (Instituto de Biologia Experimental e

Tecnológica, Universidade Nova de Lisboa, Portugal) for providing the antigen used in the ELISA and in the RBD competition assay.

## Author Contributions

**Conceptualization:** Diana Lousa, Cláudio M. Soares, Maria João Amorim.

**Data curation:** Marta Alenquer, Diana Lousa, Ricardo B. Leite, Jingtao Lilue, Maria João Amorim.

**Formal analysis:** Marta Alenquer, Filipe Ferreira, Diana Lousa, Marie-Louise Bergman, Juliana Gonçalves, Jocelyne Demengeot, Ricardo B. Leite, Jingtao Lilue, Zemin Ning, Helena Soares, Maria João Amorim.

**Funding acquisition:** Maria João Amorim.

**Investigation:** Marta Alenquer, Filipe Ferreira, Diana Lousa, Mariana Valério, Mónica Medina-Lopes, Marie-Louise Bergman, Juliana Gonçalves, Ricardo B. Leite, Jingtao Lilue, Zemin Ning, Helena Soares, Cláudio M. Soares, Maria João Amorim.

**Methodology:** Marta Alenquer, Filipe Ferreira, Diana Lousa, Mariana Valério, Mónica Medina-Lopes, Marie-Louise Bergman, Juliana Gonçalves, Maria João Amorim.

**Project administration:** Maria João Amorim.

**Resources:** Jocelyne Demengeot, Ricardo B. Leite, Carlos Penha-Gonçalves, Helena Soares, Cláudio M. Soares.

**Supervision:** Cláudio M. Soares, Maria João Amorim.

**Validation:** Marta Alenquer, Maria João Amorim.

**Writing – original draft:** Marta Alenquer, Diana Lousa, Mariana Valério, Ricardo B. Leite, Cláudio M. Soares, Maria João Amorim.

**Writing – review & editing:** Marta Alenquer, Jocelyne Demengeot, Zemin Ning, Helena Soares, Maria João Amorim.

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
