## [Decision Letter · Decision Letter 0]

17 Jun 2021

Dear Dr. Amorim,

Thank you very much for submitting your manuscript "Amino acids 484 and 494 of SARS-CoV-2 spike are hotspots of immune evasion affecting antibody but not ACE2 binding" for consideration at PLOS Pathogens. As with all papers reviewed by the journal, your manuscript was reviewed by members of the editorial board and by several independent reviewers. In light of the reviews (below this email), we would like to invite the resubmission of a significantly-revised version that takes into account the reviewers' comments.

Both Reviewers made suggestions on how to improve the study. In particular, Reviewer #2 felt the paper should be more streamlined, with consolidation of data and focusing of the writing.

We cannot make any decision about publication until we have seen the revised manuscript and your response to the reviewers' comments. Your revised manuscript is also likely to be sent to reviewers for further evaluation.

Sincerely,

Michael Diamond

Section Editor

PLOS Pathogens

Kasturi Haldar

Editor-in-Chief

PLOS Pathogens

orcid.org/0000-0001-5065-158X

Michael Malim

Editor-in-Chief

PLOS Pathogens

orcid.org/0000-0002-7699-2064

Reviewer's Responses to Questions

**Part I - Summary**

Reviewer #1: SARS-CoV-2 variants which escape antibody neutralization are of significant concern as previous studies have demonstrated that the current vaccines induce humoral immunity with less neutralization capacity against these variants. Here, the authors use a lentiviral spike-pseudotyped reporter viral particle to characterize polyclonal neutralization against the most prevalent variants of concern. Additionally the authors computationally predict, based off ACE2 and antibody binding, the amino acids most tolerable for mutations that can avoid antibody binding without effecting ACE2 binding. With this strategy, the authors have identified amino acid position 484 and 494 as likely to emerge in variant strains. These two amino acid sites have previously been characterized as hotspots for antibody escape mutants. The authors combine these specific amino acid mutations to quantify synergistic effect on neutralizing antibody responses.

Overall, this study recapitulates many of the previous results demonstrating mutational hotspots that give rise to viral mutants able to escape antibody neutralization. The unique aspect of this study is combining multiple mutations into a single reporter viral constructs to characterize the synergistic effect of specific amino acid mutations on serum neutralization capacity. Through these efforts, the authors identify that mutations in the amino acid sites 484 and 494 as the most likely sites for synergistic mutations that escape antibody neutralization. This manuscript is very clearly written and the results support the conclusions. Below is specific, minor critique.

Reviewer #2: The study from Alenquer et al. investigates the interplay between SARS-COV-2 variants and neutralizing antibodies/ spike protein. The manuscript presents three main topics: 1) analysis of SARS-COV-2 variants throughout the world, resulting in identification of a panel of emerging variants of concern with mutations in the spike protein. 2) Creation of a panel of spike-pseudotyped viruses expressing wild type and mutant spike proteins, which are then used in neutralizing antibody assays with convalescent and vaccine-immune sera to assess sensitivity to neutralization. 3) use of existing virus-spike and virus-antibody structures to determine spike protein residues frequently engaged with antibodies and/or ACE-2 receptor. One conclusion of the study is the identification of spike mutants E484K and S494 as positions that engage with neutralizing antibodies, but not ACE-2, and are thus prone to immune escape.

**Part II – Major Issues: Key Experiments Required for Acceptance**

Reviewer #1: (No Response)

Reviewer #2: Overall, the paper is extremely thorough, but lacks focus and is too long. Many of the variants studied (B.1.1.7, B.1.351) were already known, and the descriptions of the variants in supplementary Table S1 indicate that neutralization sensitivity and/or ACE-2 binding has been studied already. The sequence analysis presented is very detailed, and perhaps beyond the scope of the study. The first paragraph of the results contains too much background information, and the geography/prevalence of the individual variants is not particularly useful to the reader and could be presented more succinctly in table form. Furthermore, the numerous structures of the variants, particularly Figures S3-S5 (with individual side chains) are unnecessary and the figure legends for these figures contain too much guessing regarding the functions of the individual amino acids. The neutralization assay data is rigorous, but the large number of figures and supplementary figures is difficult to follow. The major conclusions of the paper are based on published structural data (available spike-mAb & spike-ACE-2 complexes). While the title of the paper emphasizes spike amino acids 484 and 494, this has little to do with the neutralization data presented in the results section. Overall, the authors have presented a detailed analysis of viral variants, neutralization data for a large panel of spike variants, and an analysis of the current structural data regarding spike residues targeted by neutralizing antibodies versus ACE-2. However, these results are not tied together in a cohesive story.

**Part III – Minor Issues: Editorial and Data Presentation Modifications**

Reviewer #1: Human Sera: Do the authors know the strain of the virus infecting patients from which sera was isolated? If the strain is not conclusively known, can the authors speculate on the strain based off the dominant circulating strains in Portugal at the time of blood draw? The infecting strain will likely effect the specificity and target of the humoral immune response.

Line 63-64. The cascade of spike processing is a little off here. S1/S2 is cleaved by the host protease furin during viral egress. The S2’ is cleaved after viral attachment by host protease TMPRSS2 on the cell surface. PMID: 32362314

Line 321-327: The spike-pseudotyped lentiviral GFP reporter particle is not novel and has been developed by others. (Summarized in PMID: 32384820)

Line 114: skype -> spike

Line 140: SA (South Africa)

Line 287: covalence -> convalescent

Reviewer #2: (No Response)

PLOS authors have the option to publish the peer review history of their article (what does this mean?). If published, this will include your full peer review and any attached files.

Reviewer #1: No

Reviewer #2: No
---

## [Editor Report · Decision Letter 1]

30 Jun 2021

Dear Dr. Amorim,

We are pleased to inform you that your manuscript 'Signatures in SARS-CoV-2 spike protein conferring escape to neutralizing antibodies' has been provisionally accepted for publication in PLOS Pathogens.

Best regards,

Michael S. Diamond

Section Editor

PLOS Pathogens

Michael Diamond

Section Editor

PLOS Pathogens

Kasturi Haldar

Editor-in-Chief

PLOS Pathogens

orcid.org/0000-0001-5065-158X

Michael Malim

Editor-in-Chief

PLOS Pathogens

orcid.org/0000-0002-7699-2064
---

## [Editor Report · Acceptance letter]

14 Jul 2021

Dear Dr. Amorim,

We are delighted to inform you that your manuscript, "Signatures in SARS-CoV-2 spike protein conferring escape to neutralizing antibodies," has been formally accepted for publication in PLOS Pathogens.

Best regards,

Kasturi Haldar

Editor-in-Chief

PLOS Pathogens

orcid.org/0000-0001-5065-158X

Michael Malim

Editor-in-Chief

PLOS Pathogens

orcid.org/0000-0002-7699-2064